# Synthesis of Substituted 1,2,4-Triazole-3-Thione Nucleosides Using *E. coli* Purine Nucleoside Phosphorylase

**DOI:** 10.3390/biom14070745

**Published:** 2024-06-24

**Authors:** Ilya V. Fateev, Sobirdjan A. Sasmakov, Jaloliddin M. Abdurakhmanov, Abdukhakim A. Ziyaev, Shukhrat Sh. Khasanov, Farkhod B. Eshboev, Oybek N. Ashirov, Valeriya D. Frolova, Barbara Z. Eletskaya, Olga S. Smirnova, Maria Ya. Berzina, Alexandra O. Arnautova, Yulia A. Abramchik, Maria A. Kostromina, Alexey L. Kayushin, Konstantin V. Antonov, Alexander S. Paramonov, Valeria L. Andronova, Georgiy A. Galegov, Roman S. Esipov, Shakhnoz S. Azimova, Anatoly I. Miroshnikov, Irina D. Konstantinova

**Affiliations:** 1Shemyakin and Ovchinnikov Institute of Bioorganic Chemistry, Russian Academy of Sciences, Miklukho-Maklaya St. 16/10, 117997 Moscow, Russia; lera1974@icloud.com (V.D.F.); fraubarusya@gmail.com (B.Z.E.); gescheites@gmail.com (O.S.S.); berzina_maria@mail.ru (M.Y.B.); 8818818@mail.ru (A.O.A.); ugama@yandex.ru (Y.A.A.); kostromasha@mail.ru (M.A.K.); kayushin.alexej@yandex.ru (A.L.K.); antonov.kant@yandex.ru (K.V.A.); a.s.paramonov@gmail.com (A.S.P.); esipov@ibch.ru (R.S.E.); aiv@mail.ibch.ru (A.I.M.); kid1968@yandex.ru (I.D.K.); 2Acad. S.Yu. Yunusov Institute of the Chemistry of Plant Substances, Academy of Sciences of the Republic of Uzbekistan, Mirzo Ulugbek Str. 77, 100170 Tashkent, Uzbekistan; jaloliddin0919@mail.ru (J.M.A.); aziyaev05@rambler.ru (A.A.Z.); sh.sh.hasanov@mail.ru (Sh.Sh.K.); farkhod.eshboev@gmail.com (F.B.E.); oyashirov1991@gmail.com (O.N.A.); genlab_icps@yahoo.com (S.S.A.); 3D. I. Ivanovsky Institute of Virology (N. F. Gamaleya Research Center of Epidemiology and Microbiology, Ministry of Healthcare of the Russian Federation), Gamaleya St. 18, 123098 Moscow, Russia; andronova.vl@yandex.ru (V.L.A.); kid196892@gmail.com (G.A.G.)

**Keywords:** nucleoside phosphorylases, nucleosides, ribavirin, antiviral activity

## Abstract

1,2,4-Triazole derivatives have a wide range of biological activities. The most well-known drug that contains 1,2,4-triazole as part of its structure is the nucleoside analogue ribavirin, an antiviral drug. Finding new nucleosides based on 1,2,4-triazole is a topical task. The aim of this study was to synthesize ribosides and deoxyribosides of 1,2,4-triazole-3-thione derivatives and test their antiviral activity against herpes simplex viruses. Three compounds from a series of synthesized mono- and disubstituted 1,2,4-triazole-3-thione derivatives were found to be substrates for *E. coli* purine nucleoside phosphorylase. Of six prepared nucleosides, the riboside and deoxyriboside of 3-phenacylthio-1,2,4-triazole were obtained at good yields. The yields of the disubstituted 1,2,4-triazol-3-thiones were low due to the effect of bulky substituents at the C3 and C5 positions on the selectivity of enzymatic glycosylation for one particular nitrogen atom in the triazole ring. The results of cytotoxic and antiviral studies on acyclovir-sensitive wild-type strain HSV-1/L2(TK+) and acyclovir-resistant strain (HSV-1/L2/R^ACV^) in *Vero E6* cell culture showed that the incorporation of a thiobutyl substituent into the C5 position of 3-phenyl-1,2,4-triazole results in a significant increase in the cytotoxicity of the base and antiviral activity. The highest antiviral activity was observed in the 3-phenacylthio-1-(β-*D*-ribofuranosyl)-1,2,4-triazole and 5-butylthio-1-(2-deoxy-β-*D*-ribofuranosyl)-3-phenyl-1,2,4-triazole nucleosides, with their selectivity indexes being significantly higher than that of ribavirin. It was also found that with the increasing lipophilicity of the nucleosides, the activity and toxicity of the tested compounds increased.

## 1. Introduction

Ribavirin is a broad-spectrum antiviral drug that is effective against RNA and DNA viruses, including influenza A and B, respiratory syncytial virus, cytomegalovirus, herpes simplex virus 1 and 2, and herpesvirus 6 [1,2]. Among all synthesized 1,2,4-triazole-3-carboxamide nucleosides, only ribavirin is still used in monotherapy and combination regimens for treating viral hepatitis C, respiratory syncytial virus infection, and hantavirus hemorrhagic fever with renal syndrome [3,4,5]. However, ribavirin has serious drawbacks, including teratogenicity, mutagenicity, and hemolytic activity. This has led to a search for compounds with a better therapeutic index among its structural analogues [6,7,8]. For this reason, the search for new nucleosides based on 1,2,4-triazole is still relevant.

Currently, ribavirin analogues with substituents on the heterocyclic base are being synthesized and studied for their antitumor and antiviral activities [9,10]. Thio derivatives of 1,2,4-triazoles are gaining increased importance in medicinal chemistry due to their antitumor, antimicrobial, antioxidant and anti-inflammatory properties [11]. Previously synthesized 1,2,4-triazole-3-thiones with bulky aromatic substituents showed activity against herpes simplex virus and Coxsackie B4 virus [12]. The incorporation of the thione group at the C3 or C5 position of triazole results in increased biological activity associated with the triazole moiety. The preparation of nucleoside analogues of these compounds has the potential to improve their solubility and bioavailability while maintaining their biological activity [13,14].

Another promising approach to the synthesis of modified nucleosides is the incorporation of an aromatic substituent into the triazole ring. Alkynyltriazole ribonucleosides have shown activity against herpes simplex virus while not exhibiting significant toxicity against various cell lines. However, the removal of the ethynyl group caused a loss of activity [15]. Other 1,2,4-triazole riboside derivatives have also shown antitumor activity, including against gemcitabine-resistant cell lines [16].

1,2,4-Triazole-3-carboxamide nucleosides are isosteric analogues of guanosine and are capable of being integrated into nucleic acid metabolism through the action of cellular kinases. This process significantly alters the cellular metabolic pathway of triazole nucleosides compared to that of their parent base [17]. Therefore, it is important to obtain information not only about the activity of substituted triazole base, but also to determine the activity of the 1,2,4-triazole nucleosides.

Currently, various enzymes are used to produce nucleoside analogues. Enzymatic synthesis increases the efficiency of the synthesis process and simplifies the purification procedure [18,19,20].

Based on our data, 1,2,4-triazole derivatives with bulky alkoxymethyl substituents at the C5 position have already proven to be non-standard hydrophobic substrates for *E. coli* purine nucleoside phosphorylase (PNP) in the synthesis of nucleosides [21]. Despite the apparent steric hindrance, bacterial PNPs can be used as a catalyst for the synthesis of nucleosides [22].

In this work, we synthesized mono- and disubstituted 1,2,4-triazole-3-thiones and glycosylated them into ribosides and 2-deoxyribosides using *E. coli* purine nucleoside phosphorylase. We compared the activity of these triazole bases and corresponding nucleosides against the herpes simplex virus model. We combine two approaches to the synthesis of new ribavirin analogues: the introduction of thionic (-SR) and bulky aromatic substituents at the C3 and C5 positions. These modifications can significantly change the bioavailability and biological activity of 1,2,4-triazole nucleosides.

## 2. Materials and Methods

### 2.1. General Procedures

Materials were obtained from commercial suppliers and used without any purification unless otherwise noted.

Column chromatography was performed using C18-reversed-phase silica gel (Fluka, Buchs, Switzerland). High-performance liquid chromatography (HPLC) was performed with the Waters (Waters Corporation, Milford, MA, USA) system (Waters 1525, Waters 2489, Breeze 2) using Nova Pak C18, 4.6 × 150 mm column, eluent A—0.1% TFA/H_2_O, eluent B—70% MeCN/0.1% TFA/H_2_O, flow rate 1 mL/min, detection at 230 and 254 nm. Gradient: from 0 to 30% or from 0 to 100% eluent B, 20 min.

NMR spectra were recorded on Bruker Avance II 700 spectrometer (Bruker BioSpin, Rheinstetten, Germany) in DMSO-*d6* at 30 °C, Unity 400 Plus spectrometer (Varian, Palo Alto, CA, USA) in DMSO-*d6*+CCl_4_ at 20–25 °C, and JNM-ECZ600R spectrometer (JEOL, Akishima, Japan) in DMSO-*d6*+CCl_4_ or CDCl_3_+CD_3_OD at 20–25 °C. Chemical shifts in ppm (δ) were measured relative to the residual solvent signals as internal standards (2.50 for DMSO-*d6*). Coupling constants (*J*) are expressed in Hz.

The NMR spectra are presented in the Appendix A.

IR spectra were recorded on an FT-IR/NIR Spectrum 3 spectrometer (Perkin Elmer, Shelton, CT, USA) using an ATR system.

Herpes simplex virus type 1 (HSV-1) strain L2 was obtained from the State’s collection of viruses at FSBI «N. F. Gamaleya National Research Centre for Epidemiology and Microbiology, Russian Ministry of Health, Russia» («D. I. Ivanovsky Institute of Virology» subdivision).

LogP was calculated using the ChemDraw Ultra v12.0 software.

### 2.2. Expression and Purification of Recombinant E. coli PNP and UP

Plasmid vectors containing the genes encoding *E. coli* purine nucleoside phosphorylase (PNP) and *E. coli* uridine phosphorylase (UP) constructed in previous research were used for the transformation of competent *E. coli* ER2566 (New England Biolabs, Ipswich, MA, USA) strain cells [23]. The resulting producer strains were cultivated at 37 °C in Luria–Bertani medium (10 g tryptone, 5 g yeast extract, 10 g NaCl per 1 L) with 100 µg/mL ampicillin. Upon reaching the optical density of A_595_ = 0.8, the cell cultures were supplemented with 0.4 mM of IPTG, followed by further cultivation for 4 h at 37 °C, which allowed us to obtain PNP and UP in the soluble form.

The cell biomass was harvested through centrifugation and resuspended (1:10 *w*/*v*) in buffer containing 50 mM Tris-HCl pH 8.0, 5 mM EDTA, 1 mM benzylamine hydrochloride and 1 mM PMSF. The cell suspension was subjected to ultrasonic disintegration and centrifugation to remove the cell debris. The resulting supernatant was applied onto a column packed with Q Sepharose FF sorbent (Cytiva, Washington, DC, USA), equilibrated with buffer containing 50 mM Tris-HCl pH 8.0, 5 mM EDTA. The proteins were eluted with a linear gradient 0–500 mM NaCl. The fractions containing the desired enzymes were pooled, concentrated on the YM-30 membrane (Millipore, Burlington, MA, USA) and applied onto columns packed with Superdex 200 (GE Healthcare, Chicago, IL, USA) sorbent, and the final buffer was pre-equilibrated: 50 mM Tris-HCl pH 8.0, 200 mM NaCl. 0.04% NaN_3_. Fractions containing the target enzymes were pooled and concentrated on the YM-30 membrane (Merck Millipore, Burlington, MA, USA) to a concentration of approximately 30 mg/mL. The resulting aliquots of the enzyme’s solution were stored at −80 °C for further experiments. We have measured the protein concentration according to the Bradford assay and determined the enzyme purity through SDS PAGE [24,25].

### 2.3. Enzymatic Reactions

Each reaction mixture (0.25 mL, pH 7.0) contained 1 mM of the tested heterocyclic base, 2 mM 2′-deoxyuridine, 10 mM KH_2_PO_4_, and *Escherichia coli* purine nucleoside phosphorylase (5.6 units) and uridine phosphorylase (1.7 units). An activity unit is the amount (in µmol) of natural product (hypoxanthine or uracil) formed per minute. The reaction mixtures were incubated at 50 °C. Substrate and product quantities were determined using HPLC.

### 2.4. Heterocyclic Base Synthesis

#### 2.4.1. Synthesis of 1,2,4-Triazole-3-thione (**3**)

A mixture of 22.5 g (0.25 mol) thiosemicarbazide (**1**) and 20 mL (0.5 mol) formamide was heated for 30 min at 110–120 °C. The resulting clear, pale green solution was poured onto a porcelain cup and left overnight; the precipitated white crystals were filtered, washed with water and hexane, and dried in air. Yield: 21.2 g (85%), mp 223–225 °C.

#### 2.4.2. Synthesis of 5-Phenyl-1,2,4-triazole-3-thione (**4**)

Benzoylthiosemicarbazide (**2**) (1.95 g, 0.01 mol) was dissolved in 20 mL 20% sodium hydroxide. The solution was heated for 3 h at 90–95 °C, then the reaction mixture was cooled to room temperature and hydrochloric acid was added in a neutral/slightly acidic environment. The resulting white slurry was filtered, washed with cold water, dried in air. Yield: 1.49 g (84%), mp 253–255 °C.

^1^H NMR (400 MHz, DMSO-*d6*+CCl_4_): 13.62 (s, 1H, NH triazole), 13.43 (s, 1H, NH triazole), 7.87 (m, 2H, ArH H2 and H6 phenyl), 7.39 (m, 3H, ArH H3, H4 and H5 phenyl).

^13^C NMR (100 MHz, DMSO-*d6*+CCl_4_): 167.39 (C-3 triazole), 150.45 (C-5 triazole), 130.43 (C-4 phenyl), 129.06 (C-3 and C-5 phenyl), 126.24 (C-1 phenyl), 126.08 (C-2 and C-6 phenyl).

#### 2.4.3. Synthesis of 3-Phenacylthio-1,2,4-triazole (**5**)

1,2,4-Triazole-3-thione (**3**) (0.5 g, 0.005 mol), phenacyl bromide (0.99 g, 0.005 mol), and potassium carbonate (0.68 g, 0.005 mol) were mixed in dry acetone at room temperature. After evaporating the solvent, the residue was washed several times with water, and then dried in the air, yielding beige powder. Yield: 0.98 g (90%), mp 114–115 °C.

IR spectrum (KBr, ν, cm^−1^): 1698 (C=O).

^1^H NMR (600 MHz, CDCl_3_+CD_3_OD): 8.01(s, 1H, H5 triazole), 7.92 (m, 2H, ArH H2 and H6 phenacyl), 7.53 (m, 1H, ArH H4 phenacyl), 7.41 (m, 2H, ArH H3 and H5 phenacyl), 4.64 (s, 2H, CH_2_-S).

^13^C NMR (150 MHz, CDCl_3_+CD_3_OD): 194.00 (C=O), 135.29 (C-1 phenacyl), 133.93 (C-4 phenacyl), 128.83 (C-2 and C-6 phenacyl), 128.54 (C-3 and C-5 phenacyl), 40.14 (CH_2_-S).

#### 2.4.4. Synthesis of 3-Phenacylthio-5-phenyl-1,2,4-triazole (**6**)

A mixture of 0.88 g (0.005 mol) 5-phenyl-1,2,4-triazole-3-thione (**4**), 0.99 g (0.005 mol) phenacyl bromide, and 0.68 g (0.005 mol) potassium carbonate was heated in dry acetone. After evaporating the solvent, the residue was washed alternately with water, sodium hydroxide solution and water again until a neutral solution was obtained. The obtained white powder was dried in air, and then re-crystallized from ethanol, yielding white crystals. Yield: 1.28 g (87%), mp 145–146 °C.

IR spectrum (KBr, ν, cm^−1^): 1685 (C=O).

^1^H NMR (600 MHz, DMSO-*d6*+CCl_4_): 14.25 (br. s, 1H, NH triazole), 8.01 (d, 2H, *J* = 7.3, ArH H2 and H6 phenyl), 7.90 (d, 2H, *J* = 7.3, ArH H2 and H6 phenacyl), 7.59 (t, 1H, *J* = 7.3, ArH H4 phenacyl), 7.49 (t, 2H, *J* = 7.6, ArH H3 and H5 phenacyl), 7.38 (br. s, 3H, ArH H3, H4 and H5 phenyl), 4.74 (br. s, 2H, CH_2_-S).

^13^C NMR (150 MHz, DMSO-*d6*+CCl_4_): 193.03 (C=O), 158.65 (C-5 triazole), 154.83 (C-3 triazole), 135.49 (C-1 phenacyl), 132.94 (C-4 phenyl and C-4 phenacyl), 129.47 (C-1 phenyl), 128.28 (C-3 and C-5 phenyl), 128.107 (C-2 and C-6 phenacyl), 126.83 (C-2 and C-6 phenyl), 125.75 (C-3 and C-5 phenacyl), 38.69 (CH_2_-S).

#### 2.4.5. Synthesis of 3-Butylthio-5-phenyl-1,2,4-triazole (**7**)

A mixture of 1.76 g (0.01 mol) 5-phenyl-1,2,4-triazole-3-thione (**4**), 1.12 mL (0.01 mol) butyl iodide, and 1.36 g (0.01 mol) potassium carbonate was heated in dry acetone. After evaporating the solvent, the residue was washed alternately with water, sodium hydroxide solution and water again until a neutral solution was obtained. The product was dried in the air, yielding powder with a pale milk color. Yield 2.23 g (96%), mp 96–97 °C.

^1^H NMR (400 MHz, DMSO-*d6*+CCl_4_): 14.04 (br. s, 1H, NH triazole), 7.98 (d, 2H, *J* = 7.1, ArH H2 and H6 phenyl), 7.42 (t, 2H, *J* = 7.4, ArH H3 and H5 phenyl), 7.37 (t, 1H, *J* = 7.1, ArH H4 phenyl), 3.14 (t, 2H, *J* = 7.3, S-CH_2_-CH_2_-CH_2_-CH_3_), 1.71 (quintet, 2H, *J* = 7.2, S-CH_2_-CH_2_-CH_2_-CH_3_), 1.48 (sextet, 2H, *J* = 7.4, S-CH_2_-CH_2_-CH_2_-CH_3_), 0.95 (t, 3H, *J* = 7.4, S-CH_2_-CH_2_-CH_2_-CH_3_).

^13^C NMR (100 MHz, DMSO-*d6*+CCl_4_): 159.39 (C-3 triazole), 154.77 (C-5 triazole), 128.99 (C-4 phenyl), 128.18 (C-3, C-5 phenyl), 125.73 (C-1, C-2, C-6 phenyl), 31.33 (S-CH_2_-CH_2_-CH_2_-CH_3_), 31.09 (S-CH_2_-CH_2_-CH_2_-CH_3_), 21.19 (S-CH_2_-CH_2_-CH_2_-CH_3_), 13.29 (S-CH_2_-CH_2_-CH_2_-CH_3_).

### 2.5. Nucleoside Synthesis

#### 2.5.1. Synthesis of 3-Phenacylthio-1-(β-*D*-ribofuranosyl)-1,2,4-triazole (**8**)

3-Phenacylthio-1,2,4-triazole (**5**) (0.036 g, 0.164 mmol), uridine (0.1 g, 0.41 mmol), and KH_2_PO_4_ (0.056 g, 0.412 mmol) were dissolved in 41 mL of water and neutralized by 5N potassium hydroxide. Volumes of 175 µL *E. coli* UP (298 units) and 219 µL *E. coli* PNP (306 units) were added to the reaction mixture. The reaction was incubated at 50 °C for 23 h. The reaction progress was monitored by HPLC. The mixture was concentrated in vacuo, the solution was placed on the column [octadecyl–Si 100 polyol (0.03 mm); 20 × 150 mm], and riboside **8** was eluted with EtOH (8%) in water. The product was lyophilized. Yield: 55.3 mg (89%); purity 99% (HPLC).

^1^H NMR (700 MHz): δ = 8.72 (s, 1H, H5 triazole), 8.02 (d, *J* = 9.2, 2H, ArH H2 and H6 phenacyl), 7.68 (m, 1H, ArH H4 phenacyl), 7.56 (m, 2H, ArH H3 and H5 phenacyl), 5.68 (d, *J* = 4.0, 1H, H-1′), 5.47 (d, *J* = 5.7, 1H, OH-2′), 5.09 (d, *J* = 5.5, 1H, OH-3′), 4.81 (t, *J* = 5.6, 1H, OH-5′), 4.80 (br. s, 2H, CH_2_-S), 4.25 (dd, J1 = 9.8, J2 = 3.9, 1H, H-2′), 4.04 (dd, J1 = 5.5, J2 = 9.9, 1H, H-3′), 3.88 (m, 1H, H-4′), 3.53 (m, 1H, H-5′a), 3.42 (m, 1H, H-5′b).

^13^C NMR (176 MHz): δ = 194.20 (C=O), 159.90 (C-3 triazole), 145.89 (C-5 triazole), 136.07 (C-1 phenacyl), 133.99 (C-4 phenacyl), 129.24 (C-3 and C-5 phenacyl), 128.76 (C-2 and C-6 phenacyl), 91.72 (C-1′), 85.82 (C-4′), 74.59 (C-2′), 70.60 (C-3′), 61.98 (C-5′), 39.34 (CH_2_-S).

^15^N NMR (71 MHz): δ = 280.5 (N-2), 250.5 (N-4), 229.4 (N-1).

#### 2.5.2. Synthesis of 1-(2-Deoxy-β-*D*-ribofuranosyl)-3-phenacylthio-1,2,4-triazole (**9**)

3-Phenacylthio-1,2,4-triazole (**5**) (0.036 g, 0.164 mmol), 2′-deoxyuridine (0.112 g, 0.459 mmol), and KH_2_PO_4_ (0.056 g, 0.412 mmol) were dissolved in 41 mL of water and neutralized by 5N potassium hydroxide. Volumes of 117 µL *E. coli* UP (199 units) and 146 µL *E. coli* PNP (204 units) were added to the reaction mixture. The reaction was incubated at 50 °C for 45 min. The reaction progress was monitored by HPLC. The mixture was concentrated in vacuo, the solution was placed on the column [octadecyl–Si 100 polyol (0.03 mm); 20 × 150 mm], and deoxyriboside 9 was eluted with EtOH (10%) in water. The product was lyophilized. Yield: 54.1 mg (96%); purity 97% (HPLC).

^1^H NMR (700 MHz): δ = 8.66 (s, 1H, H5 triazole), 8.02 (d, *J* = 8.1, 2H, ArH H2 and H6 phenacyl), 7.68 (m, 1H, ArH H4 phenacyl), 7.56 (m, 2H, ArH H3 and H5 phenacyl), 6.11 (t, *J* = 6.0, 1H, H-1′), 5.24 (br. s, 1H, OH-3′), 4.78 (s, 1H, CH_2_-S, Ha), 4.78 (s, 1H, CH_2_-S, Hb), 4.76 (br. s, 1H, OH-5′), 4.26 (m, 1H, H-3′), 3.79 (dd, J1 = 9.5, J2 = 5.1, 1H, H-4′), 3.43 (m, 1H, H-5′a), 3.37 (m, 1H, H-5′b), 2.46 (m, 1H, H-2′a), 2.22 (m, 1H, H-2′b).

^13^C NMR (176 MHz): δ = 194.23 (C=O), 159.65 (C-3 triazole), 145.59 (C-5 triazole), 136.10 (C-1 phenacyl), 133.98 (C-4 phenacyl), 129.23 (C-3 and C-5 phenacyl), 128.75 (C-2 and C-6 phenacyl), 88.58 (C-4′), 87.47 (C-1′), 70.84 (C-3′), 62.30 (C-5′), 39.55 (C-2′), 39.27 (CH_2_-S).

^15^N NMR (71 MHz): δ = 279.9 (N-2), 250.3 (N-4), 233.4 (N-1).

#### 2.5.3. Synthesis of 5-Phenacylthio-3-phenyl-1-(β-*D*-ribofuranosyl)-1,2,4-triazole (**10**)

3-Phenacylthio-5-phenyl-1,2,4-triazole (**6**) (0.05 g, 0.169 mmol), uridine (0.2 g, 0.846 mmol), and KH_2_PO_4_ (0.46 g, 3.385 mmol) were dissolved in 19 mL dimethylformamide and 320 mL water and neutralized by 5N potassium hydroxide. Volumes of 1.3 mL *E. coli* UP (2210 units) and 1.6 mL *E. coli* PNP (2240 units) were added to the reaction mixture. The reaction was incubated at 50 °C for 337 h. The reaction progress was monitored by HPLC. The mixture was concentrated in vacuo, the solution was placed on the column [octadecyl–Si 100 polyol (0.03 mm); 20 × 150 mm], and riboside **10** was eluted with EtOH (30%) in water. The product was lyophilized. Yield: 15.2 mg (20%); purity 96% (HPLC).

^1^H NMR (700 MHz): δ = 8.08 (d, *J* = 7.3, 2H, ArH H2 and H6 phenacyl), 7.83 (d, *J* = 7.3, 2H, ArH H2 and H6 phenyl), 7.71 (m, 1H, ArH H4 phenacyl), 7.59 (m, 2H, ArH H3 and H5 phenacyl), 7.42 (m, 1H, ArH H4 phenyl), 7.42 (m, 2H, ArH H3 and H5 phenyl), 5.72 (d, *J* = 4.2, 1H, H-1′), 5.55 (d, *J* = 6.0, 1H, OH-2′), 5.22 (d, *J* = 5.8, 1H, OH-3′), 5.05 (br. s, 2H, CH_2_-S), 4.79 (t, *J* = 5.8, 1H, OH-5′), 4.56 (q, *J* = 5.0, 1H, H-2′), 4.24 (q, *J* = 5.05, 1H, H-3′), 3.97 (m, 1H, H-4′), 3.61 (m, 1H, H-5′a), 3.49 (m, 1H, H-5′b).

^13^C NMR (176 MHz): δ =193.86 (C=O), 161.18 (C-3 triazole), 154.10 (C-5 triazole), 136.12 (C-1 phenacyl), 134.17 (C-4 phenacyl), 130.59 (C-1 phenyl), 130.02 (C-4 phenyl), 129.31 (C-3 and C-5 phenacyl), 129.18 (C-3 and C-5 phenyl), 128.88 (C-2 and C-6 phenacyl), 126.22 (C-2 and C-6 phenyl), 90.29 (C-1′), 86.34 (C-4′), 74.30 (C-2′), 71.17 (C-3′), 62.57 (C-5′), 41.06 (CH_2_-S).

^15^N NMR (71 MHz): δ = 285.4 (N-2), 219.3 (N-1).

#### 2.5.4. Synthesis of 1-(2-Deoxy-β-*D*-ribofuranosyl)-5-phenacylthio-3-phenyl-1,2,4-triazole (**11**)

3-Phenacylthio-5-phenyl-1,2,4-triazole (**6**) (0.05 g, 0.169 mmol), 2′-deoxyuridine (0.31 g, 1.354 mmol), and KH_2_PO_4_ (0.46 g, 3.385 mmol) were dissolved in 25 mL dimethylformamide and 314 mL water and neutralized by 5N potassium hydroxide. Volumes of 0.55 mL *E. coli* UP (935 units) and 0.85 mL *E. coli* PNP (1190 units) were added to the reaction mixture. The reaction was incubated at 50 °C for 22.5 h. The reaction progress was monitored by HPLC. The mixture was concentrated in vacuo, the solution was placed on the column [octadecyl–Si 100 polyol (0.03 mm); 20 × 150 mm], and deoxyriboside 11 was eluted with EtOH (20%) in water. The product was lyophilized. Yield: 10.5 mg (15%); purity 98% (HPLC).

^1^H NMR (700 MHz): δ = 8.07 (d, *J* = 7.5, 2H, ArH H2 and H6 phenacyl), 7.82 (m, 2H, ArH H2 and H6 phenyl), 7.71 (m, 1H, ArH H4 phenacyl), 7.59 (m, 2H, ArH H3 and H5 phenacyl), 7.42 (m, 1H, ArH H4 phenyl), 7.41 (m, 2H, ArH H3 and H5 phenyl), 6.19 (t, *J* = 5.9, 1H, H-1′), 5.33 (d, *J* = 4.3, 1H, OH-3′), 5.02 (br. s, 2H, CH_2_-S), 4.76 (t, *J* = 5.5, 1H, OH-5′), 4.47 (m, 1H, H-3′), 3.87 (m, 1H, H-4′), 3.54 (m, 1H, H-5′a), 3.42 (m, 1H, H-5′b), 2.79 (m, 1H, H-2′a), 2.34 (m, 1H, H-2′b).

^13^C NMR (176 MHz): δ = 193.94 (C=O), 160.91 (C-3 triazole), 153.42 (C-5 triazole), 136.09 (C-1 phenacyl), 134.15 (C-4 phenacyl), 130.67 (C-1 phenyl), 129.95 (C-4 phenyl), 129.29 (C-3 and C-5 phenacyl), 129.16 (C-3 and C-5 phenyl), 128.87 (C-2 and C-6 phenacyl), 126.16 (C-2 and C-6 phenyl), 88.74 (C-4′), 86.17 (C-1′), 71.25 (C-3′), 62.67 (C-5′), 41.02 (CH_2_-S) 39.09 (C-2′).

^15^N NMR (71 MHz): δ = 285.1 (N-2), 224.4 (N-1).

#### 2.5.5. Synthesis of 5-Butylthio-3-phenyl-1-(β-*D*-ribofuranosyl)-1,2,4-triazole (**12**)

3-Butylthio-5-phenyl-1,2,4-triazole (**7**) (0.045 g, 0.193 mmol), uridine (0.33 g, 1.351 mmol), and KH_2_PO_4_ (0.136 g, 1 mmol) were dissolved in 100 mL water and neutralized by 5N potassium hydroxide. Volumes of 0.13 mL *E. coli* UP (224 units) and 0.16 mL *E. coli* PNP (224 units) were added to the reaction mixture. The reaction was incubated at 50 °C for 72 h. The reaction progress was monitored by HPLC. The mixture was concentrated in vacuo, the solution was placed on the column [octadecyl–Si 100 polyol (0.03 mm); 20 × 150 mm], and riboside 12 was eluted with EtOH (50%) in water. The product was lyophilized. Yield: 13.9 mg (20%); purity 99% (HPLC).

^1^H NMR (700 MHz): δ = 7.98 (d, *J* = 7.3, 2H, ArH H2 and H6 phenyl), 7.50 (m, 2H, ArH H3 and H5 phenyl), 7.46 (m, 1H, ArH H4 phenyl), 5.68 (d, *J* = 4.4, 1H, H-1′), 5.53 (br s, 1H, OH-2′), 5.23 (br s, 1H, OH-3′), 4.77 (br s, 1H, OH-5′), 4.55 (t, *J* = 4.6, 1H, H-2′), 4.23 (t, *J* = 4.6, 1H, H-3′), 3.95 (m, 1H, H-4′), 3.59 (m, 1H, H-5′a), 3.48 (m, 1H, H-5′b), 3.30 (m, 2H, S-CH_2_-CH_2_-CH_2_-CH_3_), 1.71 (m, 2H, S-CH_2_-CH_2_-CH_2_-CH_3_), 1.43 (m, 2H, S-CH_2_-CH_2_-CH_2_-CH_3_), 0.92 (t, *J* = 7.4, 3H, S-CH_2_-CH_2_-CH_2_-CH_3_).

^13^C NMR (176 MHz): δ = 161.42 (C-1 phenyl), 154.73 (C-5 triazole), 130.77 (C-3 triazole), 130.02 (C-4 phenyl), 129.25 (C-3 and C-5 phenyl), 126.28 (C-2 and C-6 phenyl), 89.96 (C-1′), 86.32 (C-4′), 74.24 (C-2′), 71.16 (C-3′), 62.59 (C-5′), 33.11 (S-CH_2_-CH_2_-CH_2_-CH_3_), 31.60 (S-CH_2_-CH_2_-CH_2_-CH_3_), 21.51 (S-CH_2_-CH_2_-CH_2_-CH_3_), 13.85 (S-CH_2_-CH_2_-CH_2_-CH_3_).

^15^N NMR (71 MHz): δ = 284.8 (N-2), 220.5 (N-1).

#### 2.5.6. Synthesis of 5-Butylthio-1-(2-deoxy-β-*D*-ribofuranosyl)-3-phenyl-1,2,4-triazole (**13**)

3-Butylthio-5-phenyl-1,2,4-triazole (**7**) (0.0556 g, 0.238 mmol), 2′-deoxyuridine (0.325 g, 1.424 mmol), and KH_2_PO_4_ (0.17 g, 1.249 mmol) were dissolved in 135 mL water and neutralized by 5N potassium hydroxide. Volumes of 0.05 mL *E. coli* UP (84 units) and 0.06 mL *E. coli* PNP (84 units) were added to the reaction mixture. The reaction was incubated at 50 °C for 7.5 h. The reaction progress was monitored by HPLC. The mixture was concentrated in vacuo, the solution was placed on the column [octadecyl–Si 100 polyol (0.03 mm); 20 × 150 mm], and deoxyriboside 13 was eluted with EtOH (50%) in water. The product was lyophilized. Yield: 26.2 mg (32%); purity 92% (HPLC).

^1^H NMR (700 MHz): δ = 7.97 (d, *J* = 7.9, 2H, ArH H2 and H6 phenyl), 7.49 (m, 2H, ArH H3 and H5 phenyl), 7.45 (m, 1H, ArH H4 phenyl), 6.13 (t, *J* = 6.2, 1H, H-1′), 5.30 (d, *J* = 4.7, 1H, OH-3′), 4.73 (t, *J* = 5.6, 1H, OH-5′), 4.46 (m, 1H, H-3′), 3.85 (m, 1H, H-4′), 3.53 (m, 1H, H-5′a), 3.41 (m, 1H, H-5′b), 3.29 (m, 2H, S-CH_2_-CH_2_-CH_2_-CH_3_), 2.78 (m, 1H, H-2′a), 2.31 (m, 1H, H-2′a), 1.71 (m, 2H, S-CH_2_-CH_2_-CH_2_-CH_3_), 1.43 (m, 2H, S-CH_2_-CH_2_-CH_2_-CH_3_), 0.92 (t, *J* = 7.2, 3H, S-CH_2_-CH_2_-CH_2_-CH_3_).

^13^C NMR (176 MHz): δ = 161.13 (C-1 phenyl), 154.13 (C-5 triazole), 130.88 (C-3 triazole), 129.95 (C-4 phenyl), 129.24 (C-3 and C-5 phenyl), 126.23 (C-2 and C-6 phenyl), 88.68 (C-4′), 85.86 (C-1′), 71.32 (C-3′), 62.71 (C-5′), 39.01 (C-2′), 33.00 (S-CH_2_-CH_2_-CH_2_-CH_3_), 31.61 (S-CH_2_-CH_2_-CH_2_-CH_3_), 21.53 (S-CH_2_-CH_2_-CH_2_-CH_3_), 13.85 (S-CH_2_-CH_2_-CH_2_-CH_3_).

^15^N NMR (71 MHz): δ = 284.2 (N-2), 225.5 (N-1).

### 2.6. Cytotoxicity

Cytotoxicity was determined quantitatively by the trypan blue exclusion method [26,27,28]. The cells were incubated with 2-fold serial dilutions of tested compounds with the maintenance medium (Eagle medium and medium 199, mixed in a 1:1 ratio). Negative control wells contained cells with maintenance medium.

After 72 h of cell culture incubation in the presence of a compound at a known concentration, the culture medium was harvested, the cells were treated with Versene solution, and suspended in the previously removed support medium. The resulting cell suspension was combined with 1% trypan blue solution in a ratio of 1:1. The number of live and stained cells was determined using a Goryaev camera (Minimed, Bryansk, Russia). CC_50_ was defined as the drug concentration, at which 50% of cells die after 72 h.

### 2.7. Antiviral Activity

Antiviral activity in vitro was measured by the method of the cytopathic effect (CPE) inhibition assay. In 96-well plastic plates with the 24 h generated cell monolayer, 2-fold serial dilutions of the compounds were prepared. Cells infected with MOI of 0.1 PFU/cell were incubated at 37 °C in 5% CO_2_ atmosphere [26,27,28]. After 48 h, CPE in the control cells reached 95–100%. Antiviral activity was assessed by determining of IC_50_ and IC_95_.

Three controls were used: (1) Toxicity control. Uninfected cells were incubated in the presence of the drug in the same concentration range as in the experiment. (2) Virus control. Cells were infected under conditions described above, but the maintenance medium did not contain the drug. (3) Cell control. Uninfected cell cultures were incubated in the maintenance medium, which contained no drug. The number of dead cells incubated in the maintenance medium (cell control) was 4.74 ± 0.12%.

## 3. Results and Discussion

Aglycone 3-butylthio-5-phenyl-1,2,4-triazole was mentioned in patents [29,30] as a drug that was active against the kinases AURORA-A, FLT-3, and c-TAK1; the method of synthesis was not described. There is a known method for the synthesis of 3-aryl-5-alkylthio-1,2,4-triazoles exhibiting the properties of muscle relaxants, anticonvulsants, antispasmodics, and sedatives [31].

Classical methods for the cyclization of semicarbazides were used for the chemical synthesis of the series of substituted triazoles **5**–**7** (Figure 1). The synthesis of 1,2,4-triazole-3-thione (**3**) was carried out from thiosemicarbazide (**1**) and formamide. The synthesis of 5-phenyl-1,2,4-triazole-3-thione (**4**) was carried out from benzoylthiosemicarbazide (**2**) in a sodium hydroxide solution.

Substitution was then carried out at the thio group. 3-Phenacylthio-1,2,4-triazole (**5**) and 3-phenacylthio-5-phenyl-1,2,4-triazole (**6**) were obtained from compounds **3** and **4**, respectively, by reacting with phenacyl bromide in the presence of potassium carbonate in dry acetone. 3-Butylthio-5-phenyl-1,2,4-triazole (**7**) was obtained from compound **4** by reacting with butyl iodide in the presence of potassium carbonate in dry acetone.

The chemical synthesis of 3-butylthio-5-phenyl-1,2,4-triazole 2-deoxyriboside has not been described, most likely due to the difficulty of its implementation according to the classical glycosylation reaction scheme, which could result in the formation of a poorly separable mixture of α and β-anomers of the target compound. The formation of side N2 and N4 regioisomeric products of base glycosylation is also possible.

In this situation, it made sense to check whether bases **3**–**7** are substrates for *E. coli* purine nucleoside phosphorylase (PNP). The scheme of nucleoside synthesis by enzymatic transglycosilation is shown on Figure 2.

The process involves the synthesis of α-*D*-ribose-1-phosphate from uridine by uridine phosphorylase (UP). Then, the obtained α-*D*-ribose-1-phosphate and heterocyclic compound underwent enzymatic transglycosylation by *E. coli* purine nucleoside phosphorylase and resulted in the desired product with a higher yield, compared to using a single enzyme (PNP) and natural purine nucleoside as a ribose donor. This could happen due to the shift in the thermodynamic equilibrium of the pyrimidine nucleoside toward the formation of a heterocyclic base rather than a purine base [32].

The synthesis of 2-deoxyribosides in the PNP active site proceeds faster and easier compared to the synthesis of ribosides. The test reactions contained the tested heterocyclic base, 2′-deoxyuridine, and both enzymes (PNP and UP).

The experimental results are shown in Table 1. For compounds **5**, **6** and **7**, the product’s formation was observed according to the HPLC data. Compounds **3** and **4** did not prove to be substrates. 1,2,4-Triazole-3-thione (**3**) has the smallest size of all the compounds tested. The presence of only a triazole cycle ring and a thio group does not allow for binding to the enzyme’s active site. Such small compounds have not been reported to be substrates of nucleoside phosphorylases. 5-Phenyl-1,2,4-triazole-3-thione (**4**) has an additional phenyl group. Compounds containing a five-membered and a six-membered cycle ring are substrates for *E. coli* purine nucleoside phosphorylase [33]; however, no product formation was observed in the reaction with this compound.

At the next stage, the enzymatic synthesis of target nucleosides was carried out. The number of enzyme activity units for the experiment directly depended on the rate of the enzymatic reaction. For that purpose, a series of preliminary experiments were carried out. The optimal amount of enzyme was selected so that the reaction lasted from several hours (for good substrates) to several days or weeks (for weak substrates). The volume of the reaction mixture depended on the solubility of the starting substances.

The synthesis of ribosides of heterocyclic bases **5**, **6** and **7** was carried out in a reaction with uridine (Urd) and potassium dihydrogen orthophosphate in water (Figure 2). *E. coli* purine nucleoside phosphorylase and uridine phosphorylase were added, and the reaction mixture was incubated at 50 °C. The synthesis of ribosides **8**, **10**, **12** was carried out for 23, 337 and 72 h, respectively (Table 2). The products were isolated using reversed-phase column chromatography. The yield of monosubstituted 1,2,4-triazole riboside **8** was 89%. The disubstituted 1,2,4-triazoles **6** and **7** were found to be more complex substrates; nucleosides **10** and **12** were synthesized with lower conversions and more difficult product isolations, resulting in yields of 20% for both products.

As in the previous study [21], the formation rate of 2-deoxyribosides was an order of magnitude higher. The synthesis of 2-deoxyribosides of heterocyclic bases **5**, **6** and **7** was carried out in a reaction with 2′-deoxyuridine (dUrd) and potassium dihydrogen orthophosphate in water (Figure 2). *E. coli* purine nucleoside phosphorylase and uridine phosphorylase were added, and the reaction mixture was incubated at 50 °C. The synthesis of 2-deoxyribosides **9**, **11** and **13** was carried out for 0.75, 22.5 and 7.5 h, respectively (Table 2). The products were isolated using reversed-phase column chromatography. The yield of the monosubstituted 1,2,4-triazole 2-deoxyriboside **9** was 97%. The synthesis and purification of the disubstituted 1,2,4-triazole deoxyribosides of **11** and **13** were also much more difficult and the yield was 15 and 32%, respectively.

When analyzing the structure of the products using NMR spectroscopy, it was found that the attachment of the carbohydrate residue occurs at different positions of the 1,2,4-triazole (Figure 3). For the heterocyclic base **5**, the substitution occurs at the N1 position, which is the most distant from the bulky substituent. For the heterocyclic bases **6** and **7**, which have two bulky substituents, attachment occurs at the adjacent N2 position.

The position of glycosylation in the 1,2,4-triazole was determined using ^1^H,^13^C-HMBC NMR spectroscopy. The hydrogen atom at the first position of the ribose interacts with the nearest carbon atom in the triazole ring (Figure 3). Additionally, the carbon atom bonded to the thio substituent interacts with the methylene group of this substituent. Fragments of the ^1^H,^13^C-HMBC spectra are provided in the Appendix A.

Interestingly, in the presence of a carboxamide group or an aromatic substituent at the C3 position of the 1,2,4-triazole ring, glycosylation always occurs at the N1 atom of the 1,2,4-triazole cycle. Furthermore, when there is a carboxamide group at the C3 position and C5 substituent is larger than methyl (ethyl, phenyl, etc.), such a base is not a PNP substrate [21]. Glycosylation can occur both at the N1 and the neighboring N2 atoms (due to their similar reactivity) if substituents at positions C3 and C5 of the triazole differ from a highly charged carboxamide. This was confirmed in the synthesis of the thionic 1,2,4-triazole nucleosides. Thus, nucleosides **8** and **9** with an N1-glycosidic bond were synthesized from the monosubstituted base **5**. In contrast, glycosylation of the 3,5-disubstituted bases **6** and **7** proceeded at another nitrogen atom that is distant from both substituents as far as possible. Interestingly, both the thionic and the phenyl residues are quite bulky and hydrophobic. However, this did not prevent the binding of these bases with the PNP active site. Compounds with bulky and hydrophobic residues have previously been reported as substrates for bacterial PNPs [21,34].

The antiherpetic activity of the synthesized bases and nucleosides has been investigated (Table 3).

All compounds except for **4**, **5** and **8** were found to be more toxic to cells than ribavirin (Figure 4).

Heterocyclic bases **5** and **6** showed toxicity and antiviral activity at the level of the corresponding ribosides and deoxyribosides. Heterocyclic base **7** showed antiviral activity at the level of deoxyriboside **13**, but turned out to be very toxic. The toxicity of heterocyclic base **7**, when compared to the cytotoxicity of its “predecessor” **4**, was increased 62-fold when a C5-butyl substituent was introduced (91.07 vs. 5642.39 µM). Interestingly, the ribosylation of thiobutyl base **7** reduced its cytotoxicity by more than 7 times (657.49 vs. 91.07 µM), while deoxyribosylation reduced it by only 4 times.

The IC_50_ value for nucleoside **8** was found to be at the same level as ribavirin for acyclovir-sensitive strain HSV-1/L2 (+) and two times lower for the acyclovir-resistant strain (HSV-1/L2/R^ACV^). Therefore, compound **8** has a higher selectivity index compared to ribavirin. Compound **13** was found to be the most active: its IC_50_ value is 20 times lower than that of ribavirin, and its CC_50_ was 4 times lower; as a result, its SI is equal to 16.

Thus, it can be concluded that the glycosylation of mono- and disubstituted 1,2,4-triazole-3-thiones does not significantly change their cytotoxicity towards *Vero E6* cells or the level of antiviral activity. However, the exception is the butyl-thione derivative **7**. The antiviral activity of riboside **12** decreased by 6-fold compared to that of **7**, while the activity of 2′-deoxyriboside **13** increased by about a third. The IC_50_ value for base **7** was 33.48 µM, while that for nucleoside **13** was 23.44 µM.

The octanol/water partition coefficient (logP), which is a characteristic of lipophilicity, was calculated for the substances tested for antiviral activity (Appendix A). This information was then used to determine the relationship between lipophilicity and cytotoxicity, antiviral activity and the selectivity index (Appendix A).

For both heterocyclic bases and nucleosides, an increase in lipophilicity generally leads to an increase in both toxicity and antiviral activity. However, there is one exception: base **4**, which has very low toxicity and activity, does not follow the general trend. For all nucleosides except **8**, a linear decrease in the CC_50_ value is observed with increasing LogP. Additionally, for all nucleosides, a linear decrease in the IC_50_ value is observed with increasing LogP. Appendix A with the mathematical models is available in the Appendix A. Mathematical models have not been developed for heterocyclic bases due to their limited number. There is no simple relationship between the selectivity index and lipophilicity for the investigated compounds.

## 4. Conclusions

New 1,2,4-triazole-3-thione nucleosides with bulky substituents at positions 3 and 5 of the heterocyclic base were synthesized using an enzymatic transglycosylation reaction. Interestingly, *E. coli* purine nucleoside phosphorylase can accept 1,2,4-triazoles with bulky substituents as substrates, and 2′-deoxyribonucleosides were synthesized an order of magnitude faster than ribonucleosides.

The antiviral activity of triazole derivatives and their nucleosides against the herpes simplex virus type 1 of two strains has been studied. Cytotoxicity towards *Vero E6* cells was found to be higher than for the ribavirin comparison drug, with the exception of 3-phenacylthio-1-(β-*D*-ribofuranosyl)-1,2,4-triazole (**8**). Two synthesized nucleosides, 3-phenacylthio-1-(β-*D*-ribofuranosyl)-1,2,4-triazole (**8**) and 5-butylthio-1-(2-deoxy-β-*D*-ribofuranosyl)-3-phenyl-1,2,4-triazole (**13**), showed significant antiviral activity, with selectivity indexes several times higher than that of ribavirin. The antiviral activity and cytotoxicity of the tested compounds increased with increasing lipophilicity.

Next, we are going to synthesize and test new 1,2,4-triazole-3-thione derivatives for their antiviral activity on various models.

## Figures and Tables

**Figure 1 biomolecules-14-00745-f001:**
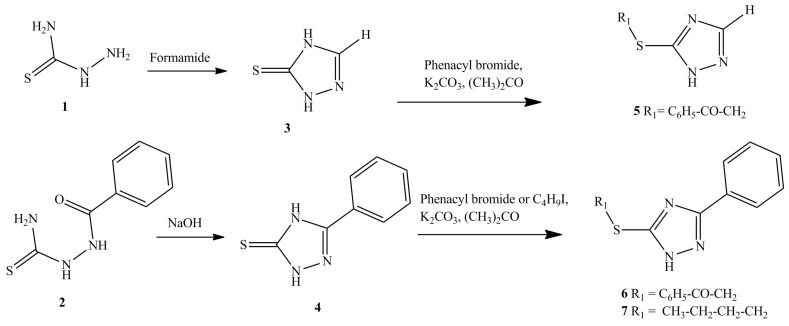
Synthesis of 3- and 5-substituted 1,2,4-triazoles.

**Figure 2 biomolecules-14-00745-f002:**
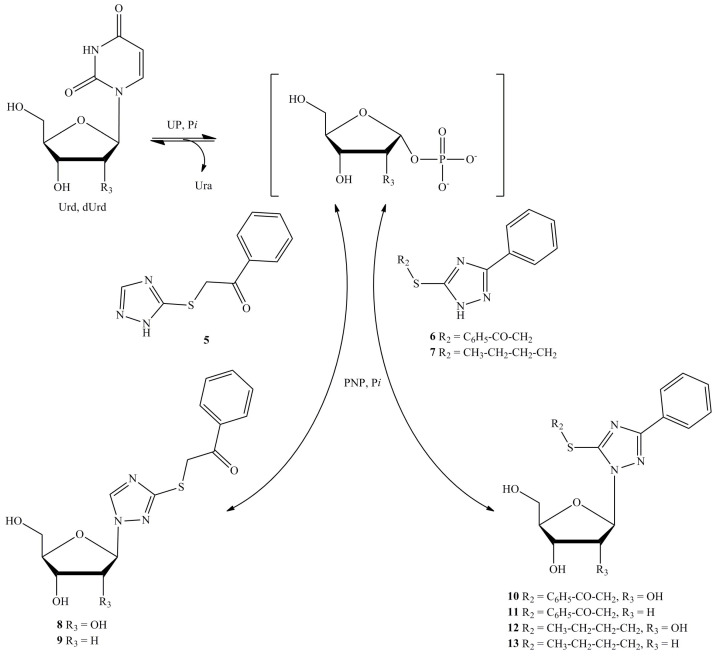
Synthesis of nucleosides **8**–**13** using nucleoside phosphorylases. PNP—*E. coli* purine nucleoside phosphorylase, UP—*E. coli* uridine nucleoside phosphorylase, Urd—uridine, dUrd—2′-deoxyuridine, Ura—uracil, Pi—inorganic phosphate.

**Figure 3 biomolecules-14-00745-f003:**
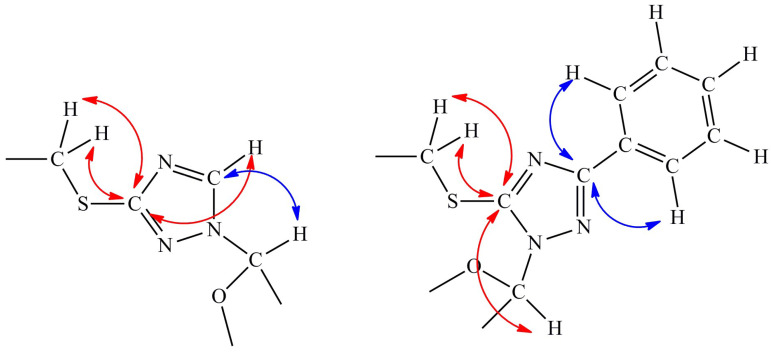
The interaction of carbon and hydrogen atoms observed in the ^1^H,^13^C-HMBC spectra for two glycosylation variants. Red arrows depict the interactions between the carbon atom bonded to the thio substituent and protons of the molecule, and blue arrows are the interactions of the other carbon atom in triazole ring.

**Figure 4 biomolecules-14-00745-f004:**
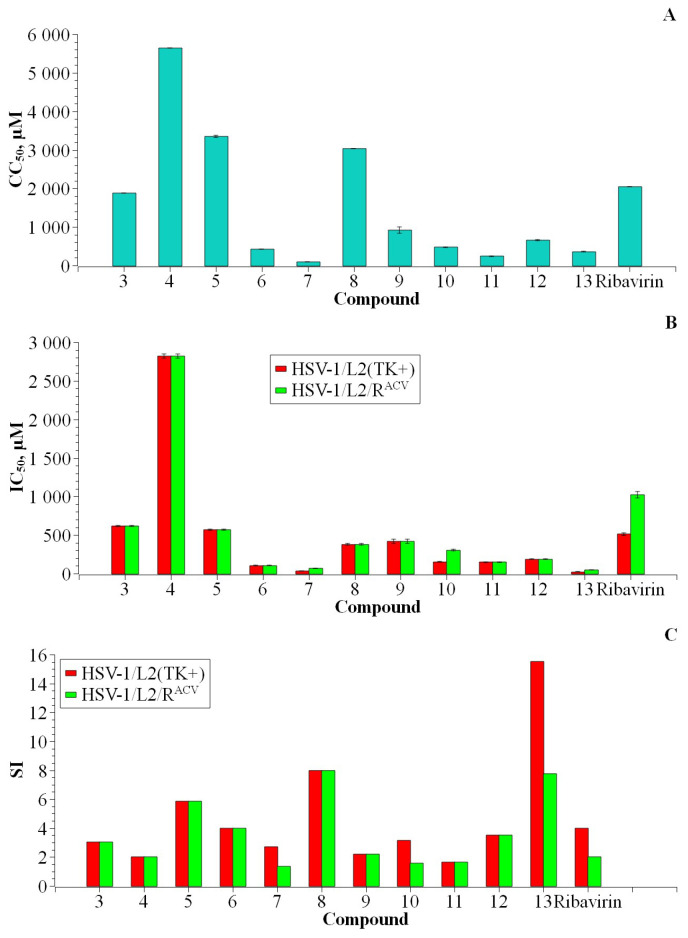
Cytotoxic properties CC_50_ (**A**), antiviral activity IC_50_ (**B**), and selectivity index SI (**C**) of compounds **3**–**13** and ribavirin.

**Table 1 biomolecules-14-00745-t001:** Product content in the test reaction mixtures for 2-deoxyribosides synthesis. Reaction conditions: volume 0.25 mL, 1 mM of the tested heterocyclic base, 2 mM 2′-deoxyuridine, 10 mM KH_2_PO_4_, pH 7.0, 5.6 units PNP, 1.7 units UP, temperature 50 °C.

Compound	Deoxyriboside Content, %	
	1 h	24 h
**3**	0	0
**4**	0	0
**5**	94	25
**6**	57	28
**7**	62	41

**Table 2 biomolecules-14-00745-t002:** Synthesis conditions and product yield.

Comp.	Comp. Mass, mg	PNP/UP, Units	Comp. to Urd (or dUrd) Molar Ratio	Vol., mL	Reaction Time, h	Base to Nucleoside Conversion, %	Yield, mg	Yield, %
**8**	36	219/298	1:2.5	41	23	89.6	55.3	89
**9**	36	204/199	1:2.8	41	0.75	99.5	54.1	97
**10**	50	2240/2210	1:5	339	337	53.3	15.2	20
**11**	50	1190/935	1:8	339	22.5	59.7	10.5	15
**12**	45	224/224	1:7	100	72	86.5	13.9	20
**13**	55.6	84/84	1:6	135	7.5	93.1	26.2	32

**Table 3 biomolecules-14-00745-t003:** Cytotoxic properties and antiviral activity of bases and nucleosides against acyclovir-sensitive wild-type strain HSV-1/L2(TK+) and acyclovir-resistant strain (HSV-1/L2/R^ACV^) in *Vero E6* cell culture.

Compound	CC_50_ ^1^, µM	IC_50_ ^2^, µM	IC_95_ ^2^, µM	SI ^3^	IC_50_, µM	IC_95_, µM	SI
		Strain HSV-1/L2(TK+)			Strain HSV-1/L2/R^ACV^		
**3**	1885.7 ± 1.7	618.0 ± 3.7	>1236.0	3	618.0 ± 3.7	>1236.0	3
**4**	>5642.4	2821.2 ± 25.4	>5642.4	>2	2821.2 ± 25.4	>5642.4	>2
**5**	3346.4 ± 26.5	570.1 ± 5.1	2280.4 ± 20.5	6	570.1 ± 5.1	2280.4 ± 20.5	6
**6**	>423.2	105.8 ± 1.2	423.2 ± 4.6	>4	105.8 ± 1.2	423.2 ± 4.6	>4
**7**	91.1 ± 0.7	33.5 ± 0.4	>67.0	3	67.0 ± 0.8	>67.0	1
**8**	>3031.3	378.9 ± 11.4	1515.7 ± 45.8	>8	378.9 ± 11.4	3031.3 ± 91.5	>8
**9**	920.5 ± 77.2	417.5 ± 25.9	834.9 ± 51.8	2	417.5 ± 25.9	834.9 ± 51.8	2
**10**	479.6 ± 6.2	152.3 ± 4.4	304.6 ± 8.8	3	304.6 ± 8.8	304.6 ± 8.8	2
**11**	244.3 ± 6.2	148.7 ± 5.1	148.7 ± 5.1	2	148.7 ± 5.1	148.7 ± 5.1	2
**12**	657.5 ± 13.9	187.5 ± 3.2	375.0 ± 6.4	4	187.5 ± 3.2	375.0 ± 6.4	4
**13**	363.7 ± 6.0	23.4 ± 0.6	93.8 ± 2.4	16	46.9 ± 1.2	93.8 ± 2.4	8
Ribavirin	>2047.5	511.9 ± 20.0	>2047.5	>4	1023.8 ± 39.9	>2047.5	>2

^1^ CC_50_—50% cytotoxic concentration of compound required to reduce the cell viability by 50%. ^2^ IC_50_ and IC_95_—50% or 95% inhibitory concentrations of compound producing 50% or 95–100% inhibition of the development of the virus-induced cytopathic effect, respectively, relative to complete cytopathic effect in infected but untreated control. Values are results for at least two independent experiments. ^3^ The selectivity index (SI) of each compound was determined as the ratio of the CC_50_ to the IC_50_. “>”—the number of dead cells does not reach 50% or 95% even when using the compound at the maximum concentration studied.

## Data Availability

The data presented in this study are available on request from the corresponding author.

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
