# Peer review of "Synthesis of Substituted 1,2,4-Triazole-3-Thione Nucleosides Using E. coli Purine Nucleoside Phosphorylase"

_biomolecules, 2024, doi:10.3390/biom14070745_

Round 1

Reviewer 1 Report

Comments and Suggestions for Authors

While the study may be interesting, in its current version it is very difficult to read and understand. It is difficult to judge the quality of the study as presented now. 

The authors are advised to:

- explain carefully the rationale for preparing the target analogues

- the advances of the enzyme-mediated synthesis described here compared to reported methods (e.g. extended substrate scope)

-the reason for combining two enzymes (UP and PNP) in the glycosidation reaction

Since the antiviral (HSV) activity of the synthesized analogues is rather poor, I do not see a reason to study the influence of logP.

Comments on the Quality of English Language

-

Author Response

Q1. Explain carefully the rationale for preparing the target analogues

The following text is added to the introduction: “We combine two approaches to the synthesis of new ribavirin analogues: the introduction of thionic (-SR) and bulky aromatic substituents at the C3 and C5 positions. These modifications can significantly change the bioavailability and biological activity of 1,2,4-triazole nucleosides.”

To study the relationship between structure and biological activity, we synthesized nucleosides with a thionic substituent but without an aromatic one. Unfortunately, we didn’t manage to synthesize 1,2,4-triazole-3-thione (3) and 5-phenyl-1,2,4-triazole-3-one (4)  nucleosides, because those bases are not substrates for E. coli PNP. So far, we have not included this text into Introduction. If you recommend, we’ll do this.

Q2. The advances of the enzyme-mediated synthesis described here compared to reported methods (e.g. extended substrate scope).

The following text is added to the introduction: “Currently, nucleoside analogues are synthesized using various enzymes, increases the efficiency of the synthesis process and simplifies the purification procedure.”

Some references are added to the paper:

Lapponia et al. (https://doi.org/10.1016/j.molcatb.2016.08.015)

Motter et al. (https://doi.org/10.1039/d3np00051f)

Q3. The reason for combining two enzymes (UP and PNP) in the glycosidation reaction.

For better understanding of the glycosilation reaction mechanism, we’ve changed the scheme of nucleoside synthesis (Fig. 2). The following text is added:

The process involves the synthesis of α-D-ribose-1-phosphate from uridine by uridine phosphorylase. This α-D-ribose-1-phosphate and heterocyclic base is then used by purine nucleoside phosphorylase to produce the desired product. This method often results in a higher yield of the desired product compared to using a single enzyme (PNP) and natural purine nucleoside as ribose donor. This is because the thermodynamic equilibrium for pyrimidine nucleoside is more strongly shifted toward the formation of a heterocyclic base than for a purine one.

For example see Kaspar et al. work (https://doi.org/10.1002/adsc.201901230).

Q4. Since the antiviral (HSV) activity of the synthesized analogues is rather poor, I do not see a reason to study the influence of logP.

We believe that exploration of the relationship of cytotoxicity and antiviral activity with LogP, even if the activity is poor, is interesting.

The English has been improved. The abstract has been corrected. The discussion of the results has been improved.

Reviewer 2 Report

Comments and Suggestions for Authors

I have completed a thorough review of the manuscript titled "Synthesis of Substituted 1,2,4-Triazole-3-thione Nucleosides Using E. coli Purine Nucleoside Phosphorylase." In this study, several 1,2,4-triazole derivatives with thio substituents were synthesized. Three of these compounds exhibited substrate activity for E. coli purine nucleoside phosphorylase, forming deoxyribosides more rapidly than ribosides. Notably, one of the synthesized nucleoside analogues demonstrated a significantly lower IC50 against herpes simplex virus type 1 compared to ribavirin. The manuscript is written in a satisfactory level of English, yet several revisions are needed to enhance its quality and readiness for publication. 

I recommend the following revisions: 

  1. Change the title of section 2.2 from “Antivirus activity” to “Antiviral activity.” 

  1. Correct "2-deoxyribose" to "2’-deoxyribose" and "2-deoxyuridine" to "2’-deoxyuridine." Ensure accuracy with these symbols “и.”

  1. In line 365 and 379, remove spaces between number and “%”. 

Additional Comments and Questions:

  • Was ionic strength taken into account when adjusting to pH 7.0 with a 5 M solution of potassium hydroxide? 

  • How was logP calculated?

  • Figures 4 and 5 are missing error bars. Is there statistical significance in the data presented? 

The relationship between lipophilicity and cytotoxicity, antiviral activity, and selectivity needs a deeper discussion. The correlation between logP and the selectivity index (SI) is not clearly established. I recommend developing a mathematical model to better understand these relationships, such as linear or logarithmic regression. Additionally, numbering the graphs for better identification of compounds is advised. 

Furthermore, Tables 3 and 4 should be moved to the supplementary material as they do not provide any new information beyond what is presented in the subsequent graphs. 

By addressing these points, the manuscript will be significantly improved and ready for publication. 

Comments on the Quality of English Language

The manuscript is written in a satisfactory level of English, yet several revisions are needed to enhance its quality and readiness for publication 

Author Response

Q1. Change the title of section 2.2 from “Antivirus activity” to “Antiviral activity.”

“Antivirus activity” was changed to “antiviral activity”.

Q2. Correct "2-deoxyribose" to "2’-deoxyribose" and "2-deoxyuridine" to "2’-deoxyuridine." Ensure accuracy with these symbols “и.”

“2-deoxyuridine” is changed to “2`-deoxyuridine”. “и” is changed to “and”.

"2-deoxyribose" is not changed to "2`-deoxyribose" because symbol ` is used for numbering of carbohydrates as part nucleosides.

Q3. In line 365 and 379, remove spaces between number and “%”.

All spaces between number and “%” were removed.

Q4. Was ionic strength taken into account when adjusting to pH 7.0 with a 5 M solution of potassium hydroxide?

No, ionic strength was not taken into account. The volume of KOH solution added is very small, so we can ignore the change of ionic strength.

Q5. How was logP calculated?

LogP was calculated using the ChemDraw Ultra v12.0 software. This information was added to Materials and Methods section.

Q6. Figures 4 and 5 are missing error bars. Is there statistical significance in the data presented?

Error bars are added into Fig. 4. Error bars in Fig. 5 will be invisible - the size of bars is less than point size.

Q7. The relationship between lipophilicity and cytotoxicity, antiviral activity, and selectivity needs a deeper discussion. The correlation between logP and the selectivity index (SI) is not clearly established. I recommend developing a mathematical model to better understand these relationships, such as linear or logarithmic regression. Additionally, numbering the graphs for better identification of compounds is advised.

The numbers of compounds have been added to Figure 5. Mathematical models (linear regressions) have been added to Supplementary Materials. The discussion of the results has been improved.

Q8. Furthermore, Tables 3 and 4 should be moved to the supplementary material as they do not provide any new information beyond what is presented in the subsequent graphs.

Table 3 contains important experimental data. This data is used in Discussion. We believe that this table shouldn’t be moved into Supplementary.

Table 4 has been moved to Supplementary Materials.

Q9. The manuscript is written in a satisfactory level of English, yet several revisions are needed to enhance its quality and readiness for publication

The English has been improved.

Reviewer 3 Report

Comments and Suggestions for Authors

The work by Fateev et al. presents the chemo-enzymatic synthesis of nucleoside derivatives of the well-known antiviral, Ribavirin. The authors report the chemical synthesis of a pool of purine bases and the use of two enzymes for the transglycosylation reaction. Moreover, this work reports the cytotoxicity of the synthesized compounds. While the concept is very interesting and appealing for the scientific community working on synthesis of nucleoside analogues, the manuscript shows an important lack of clarity. Therefore, I have mainly assessed the quality of the manuscript at this stage.

-          Regarding the writing style, the article would benefit from an English proof-reading.

-          Likewise, numerous typos are encountered, for instance “phosphoryldase.” in line 72. Please, proof-read the manuscript and ESI before re-submission.

-          References are scarce and almost 90% of the references are more than 5 years old. It is important to provide an adequate state of the art to facilitate the reading. This is especially important for such an active research topic as the biocatalytic synthesis of nucleoside analogues with pharmaceutical properties.

-          The introduction would benefit from the addition of recent references. For example: https://www.mdpi.com/2073-4344/9/4/355  in line 69. Other suitable and recent references are: https://pubs.rsc.org/en/content/articlehtml/2024/np/d3np00051f, https://www.sciencedirect.com/science/article/pii/S138111771630159X.

-          The “materials and methods” section does not provide sufficient experimental details. Please, describe briefly the production of the enzyme as the cited paper is more than 10 years old. Moreover, the reaction conditions for the biotransformations as well as the sample preparation for HPLC analysis are missing.

-          Uridine phosphorylase (UP), which is the second enzyme used in this work is not mentioned at all until the “results and discussion” section. How was it produced? Why was UP tested herein?

-          The vast majority of the text for “results and discussion” describes again the experimental methodology, which should be included in the previous section. However, discussion about the present work and highlight of its novelty is absent.

-          Table 2 requires more details as figures/tables must be stand-alone information. Why the volume is so different for the various reactions? It is difficult to compare the biocatalytic efficiency when the reaction conditions are so different among the samples.

-          Table 4 could be added to the supporting information because it is not relevant information and it might distract the attention of the central topic of this work.

To sum up, I recommend rejection of the present work and I encourage the authors to improve the quality of the manuscript for future submissions to highlight the experimental findings.

Comments on the Quality of English Language

The article would benefit from an English proof-reading. The sentences seem unconnected quite often. Grammar use is poor over the manuscript. Numerous typos are encountered, for instance “phosphoryldase.”

Author Response

Q1. Likewise, numerous typos are encountered, for instance “phosphoryldase.” in line 72. Please, proof-read the manuscript and ESI before re-submission.

Phosphoryldase was changed to phosphorylase. “2-deoxyuridine” was changed to “2`-deoxyuridine”. “и” was changed to “and”. All spaces between number and “%” were removed.

Q2. References are scarce and almost 90% of the references are more than 5 years old. It is important to provide an adequate state of the art to facilitate the reading. This is especially important for such an active research topic as the biocatalytic synthesis of nucleoside analogues with pharmaceutical properties.

Some references were added to the paper:

Kaspar et al., 2020, https://doi.org/10.1002/adsc.201901230

Cosgrove and Miller, 2022, https://doi.org/10.1080/17460441.2022.2039620

Q3. The introduction would benefit from the addition of recent references. For example: https://www.mdpi.com/2073-4344/9/4/355 in line 69. Other suitable and recent references are: https://pubs.rsc.org/en/content/articlehtml/2024/np/d3np00051f, https://www.sciencedirect.com/science/article/pii/S138111771630159X.

The references were added to the paper.

Q4. The “materials and methods” section does not provide sufficient experimental details. Please, describe briefly the production of the enzyme as the cited paper is more than 10 years old. Moreover, the reaction conditions for the biotransformations as well as the sample preparation for HPLC analysis are missing.

The enzymes were obtained using the same methods as in the cited Esipov et al. article. “Enzymatic Reactions” section with test reaction conditions was added to the paper. HPLC analysis of reaction mixtures was performed immediately after sampling, or samples have been frozen. No special sample preparation was performed.

Q5. Uridine phosphorylase (UP), which is the second enzyme used in this work is not mentioned at all until the “results and discussion” section. How was it produced? Why was UP tested herein?

For better understanding of the glycosilation reaction mechanism, we’ve changed the scheme of nucleoside synthesis (Fig. 2). The following text is added:

The process involves the synthesis of α-D-ribose-1-phosphate from uridine by uridine phosphorylase. This α-D-ribose-1-phosphate and heterocyclic base is then used by purine nucleoside phosphorylase to produce the desired product. This method often results in a higher yield of the desired product compared to using a single enzyme (PNP) and natural purine nucleoside as ribose donor. This is because the thermodynamic equilibrium for pyrimidine nucleoside is more strongly shifted toward the formation of a heterocyclic base than for a purine one.

For example see Kaspar et al. work (https://doi.org/10.1002/adsc.201901230).

Q6. The vast majority of the text for “results and discussion” describes again the experimental methodology, which should be included in the previous section. However, discussion about the present work and highlight of its novelty is absent.

The discussion of the results has been improved.

Q7. Table 2 requires more details as figures/tables must be stand-alone information. Why the volume is so different for the various reactions? It is difficult to compare the biocatalytic efficiency when the reaction conditions are so different among the samples.

The amount of enzyme activity units added to the reaction depended on the rate of synthesis. First, preliminary experiments have been carried out. The amount of enzyme was chosen so that the reaction lasted from a few hours (for good substrates) to a few days or weeks (for weak substrates). The volume of the reaction mixture depended on the solubility of the compound.

This information was added to the paper.

Q8. Table 4 could be added to the supporting information because it is not relevant information and it might distract the attention of the central topic of this work.

Table 4 has been moved to Supplementary Materials.

Q9. The article would benefit from an English proof-reading. The sentences seem unconnected quite often. Grammar use is poor over the manuscript. Numerous typos are encountered, for instance “phosphoryldase.”

The English has been improved.

Reviewer 4 Report

Comments and Suggestions for Authors

Minor points:

Q1. The abstract should be improved. It could highlight this work with the data. Also, the organization of the abstract is poor.

Q2. It should give the description and application of purine nucleoside phosphoryldase. Moreover, the corresponding citations should be added.

Q3. The organization of “Materials and Methods” should be improved.

Q4. How do the authors prepare the E. coli UP and E. coli PNP?

Q5. What’s the activity definition of purine nucleoside phosphoryldase?

Q6. For the synthesis of compounds 3-12, how do the authors add different activities of enzymes to the reaction mixture? For example, text “E. coli UP (2210 units) and E. coli PNP (2240 units)”, “E. coli UP (935 units) andE. coli PNP (1190 units)”, and “E. coli UP (224 units) and E. coli PNP (224 252 units)”.

Q7. It claims that three derivatives could be glycosylated to ribosides and 2-deoxyribosides by E. coli purine nucleoside phosphoryldase. Why does it add the E. coli UP to the reaction system?

Q8. What’s the optimal pH for the E. coli purine nucleoside phosphoryldase?

Q9. Why do not perform the purine nucleoside phosphoryldase-catalyzed reaction using buffer-solvent system? For example, the synthesis of compound 9 was conducted the following conditions “0.056 g (0.412 mmol) KH2PO4 were dissolved in 41 mL of water and neutralized by 5N potassium hydroxide”. Also, what’s function of KH2PO4?

Q10. Please check the text “13C NMR (176 MHz):”. It should be “13C NMR (175 MHz)”.

Q11. it should add the error data to Tables and Figures. Moreover, Table 4 could be moved to SI.

Q12. There are some problems with sentence structure, verb tense, and language expression. For example, do not start sentence with numbers (See, text in lines 207 and 229, etc.) The English of this manuscript must be improved before publication. It is suggested that you obtain assistance from a colleague who is well-versed in English or whose native language is English.

Comments on the Quality of English Language

Q12. There are some problems with sentence structure, verb tense, and language expression. For example, do not start sentence with numbers (See, text in lines 207 and 229, etc.) The English of this manuscript must be improved before publication. It is suggested that you obtain assistance from a colleague who is well-versed in English or whose native language is English.

Author Response

Q1. The abstract should be improved. It could highlight this work with the data. Also, the organization of the abstract is poor.

The abstract has been corrected.

Q2. It should give the description and application of purine nucleoside phosphoryldase. Moreover, the corresponding citations should be added.

Done.

Q3. The organization of “Materials and Methods” should be improved.

“Enzymatic Reactions” section with test reaction conditions was added. Synthesis section was divided to “Heterocyclic Bases Synthesis” and “Nucleosides Synthesis” sections. Each synthesis method was assigned a number.

Q4. How do the authors prepare the E. coli UP and E. coli PNP?

The reference to the article of Esipov et al. (https://doi.org/10.1006/prep.2001.1524) is included into Materials and methods. The production of recombinant E. coli nucleoside phosphorylases is described there. The enzymes used in this work were obtained using the same methods.

Q5. What’s the activity definition of purine nucleoside phosphoryldase?

The enzymatic activity was determined spectrophotometrically as described in Krenitsky et al. work (https://doi.org/10.1016/S0021-9258(17)33354-9) using uridine and inosine as substrates. Activity unit is the amount (in µmol) of product formed per minute. Information about activity units was added to “Materials and Methods” section.

Q6. For the synthesis of compounds 3-12, how do the authors add different activities of enzymes to the reaction mixture? For example, text “E. coli UP (2210 units) and E. coli PNP (2240 units)”, “E. coli UP (935 units) and E. coli PNP (1190 units)”, and “E. coli UP (224 units) and E. coli PNP (224 252 units)”.

The amount of enzyme activity units added to the reaction depended on the rate of synthesis. First, preliminary experiments have been carried out. The amount of enzyme was chosen so that the reaction lasted from a few hours (for good substrates) to a few days or weeks (for weak substrates).

This information was added to the paper.

Q7. It claims that three derivatives could be glycosylated to ribosides and 2-deoxyribosides by E. coli purine nucleoside phosphoryldase. Why does it add the E. coli UP to the reaction system?

For better understanding of the glycosilation reaction mechanism, we’ve changed the scheme of nucleoside synthesis (Fig. 2). The following text is added:

The process involves the synthesis of α-D-ribose-1-phosphate from uridine by uridine phosphorylase. This α-D-ribose-1-phosphate and heterocyclic base is then used by purine nucleoside phosphorylase to produce the desired product. This method often results in a higher yield of the desired product compared to using a single enzyme (PNP) and natural purine nucleoside as ribose donor. This is because the thermodynamic equilibrium for pyrimidine nucleoside is more strongly shifted toward the formation of a heterocyclic base than for a purine one.

For example see Kaspar et al. work (https://doi.org/10.1002/adsc.201901230).

Q8. What’s the optimal pH for the E. coli purine nucleoside phosphoryldase?

E. coli purine nucleoside phosphorylase has pH optimum in range 6-8. For example see Jensen and Nygaard work (https://doi.org/10.1111/j.1432-1033.1975.tb03925.x).

Q9. Why do not perform the purine nucleoside phosphoryldase-catalyzed reaction using buffer-solvent system? For example, the synthesis of compound 9 was conducted the following conditions “0.056 g (0.412 mmol) KH2PO4 were dissolved in 41 mL of water and neutralized by 5N potassium hydroxide”. Also, what’s function of KH2PO4?

KH2PO4 is the substrate for E. coli uridine phosphorylase that synthesizes α-D-ribose-1-phosphate. Our practice shows that reactions catalyzed by nucleoside phosphorylases proceed well in a phosphate buffer with a low KH2PO4 concentration. The use of other buffer solutions is not necessary.

Q10. Please check the text “13C NMR (176 MHz):”. It should be “13C NMR (175 MHz)”.

No, it was really 176 MHz. Please see the attachment.

Q11. it should add the error data to Tables and Figures. Moreover, Table 4 could be moved to SI.

Error data was added to Table 3 and Figure 4.

Q12. There are some problems with sentence structure, verb tense, and language expression. For example, do not start sentence with numbers (See, text in lines 207 and 229, etc.) The English of this manuscript must be improved before publication. It is suggested that you obtain assistance from a colleague who is well-versed in English or whose native language is English.

Sentences that started with numbers have been corrected. The English has been improved.

Round 2

Reviewer 1 Report

Comments and Suggestions for Authors

The rationale for the synthesize analogues remain poor. The authors write “Previously synthesized 4-(N-pyridylcarboxamido)-5-mercapto-3-substituted 1,2,4-triazoles have been described to exhibit significant anti-tubercular activity [12].” Referring to the effect of a sulfur on totally different triazoles does not make sense because these have a different mode of action.

I think the paper would become much stronger if the authors would focus on the substrate scope of their enzymatic glycosylation (explaining which substrates work and which don’t) rather than to give a medicinal twist to it.

Fig.5 should really be removed from the manuscript. There’s no use to study the SAR of inactive compounds only.

Replace Hemcitabine by gemcitabine

Comments on the Quality of English Language

Proofreading of the manuscript by a native English speaking colleague is recommended.

Author Response

Q1. The rationale for the synthesize analogues remain poor. The authors write “Previously synthesized 4-(N-pyridylcarboxamido)-5-mercapto-3-substituted 1,2,4-triazoles have been described to exhibit significant anti-tubercular activity [12].” Referring to the effect of a sulfur on totally different triazoles does not make sense because these have a different mode of action.

The sentence has been revised: “Previously synthesized 1,2,4-triazole-3-thiones with bulky aromatic substituents showed activity against herpes simplex virus and Coxsackie B4 virus.”

Reference [12] has been replaced by Küçükgüzel et al. work (https://doi.org/10.1016/j.ejmech.2014.11.033).

Q2. I think the paper would become much stronger if the authors would focus on the substrate scope of their enzymatic glycosylation (explaining which substrates work and which don’t) rather than to give a medicinal twist to it.

The following text is added:

1,2,4-Triazole-3-thione (3) has the smallest size of all compounds tested. The presence of only a triazole cycle ring and a thio group does not allow binding to the enzyme active site. Such small compounds have not been reported to be substrates of nucleoside phosphorylases. 5-Phenyl-1,2,4-triazole-3-thione (4) has an additional phenyl group. Compounds containing a five-membered and a six-membered cycle ring are substrates for E. coli purine nucleoside phosphorylase [33], however, no product formation was observed in the reaction with this compound.

New reference [33]: Khandazhinskaya et al. (https://doi.org/10.1039/d1ob01069g) is added.

Q3. Fig.5 should really be removed from the manuscript. There’s no use to study the SAR of inactive compounds only.

Figure 5 has been moved to Supplementary Materials.

Q4. Replace Hemcitabine by gemcitabine.

Hemcitabine was changed to gemcitabine.

Q5. Proofreading of the manuscript by a native English speaking colleague is recommended.

The English has been improved.

Reviewer 2 Report

Comments and Suggestions for Authors

After reviewing the second version of the manuscript entitled 'Synthesis of Substituted 1,2,4-Triazole-3-thione Nucleosides Using E. coli Purine Nucleoside Phosphorylase,' it appears suitable for publication in its current form following the authors' revisions.

Author Response

Thanks for the review.

Reviewer 3 Report

Comments and Suggestions for Authors

Despite the interesting experimental work carried out by Fateev et al., the quality of presentation of the results in this manuscript is still low and poorly understandable.  It is very difficult to assess the quality of the scientific results when the presentation is not clear. On the other hand, it is clear that the authors have improved the ESI and the scientific reults per se.

-          The sentences of the abstract (especially at the beginning) seem a list of bullet points. This should be corrected and the sentences should be connected one to another so that the text follows a story. The abstract must present, which is the problem and the current state of the art? What is the goal of this work? How have the authors addressed the problem? What is the main conclusion of the study?

-          Yet, some new references that have been added to the Reference List are not cited in the text. For example: ref 30, Kaspar et al., 2020, https://doi.org/10.1002/adsc.201901230. Maybe the reference 29 in line 376 should be ref 30?

-          Although the introduction has been improved, substantial changes are still needed. These two sentences say basically the same. Please, modify accordingly: “Currently, research is ongoing to create new nucleosides based on 1,2,4-triazole 50 that have low cytotoxicity and high antiviral activity.” “Currently, ribavirin analogues with substituents on the heterocyclic base are being synthesized and studied for their antitumor and antiviral activities [9,10].”

-          This sentences does not make sense. Please, revise: “Currently, nucleoside analogues are synthesized using various enzymes, increases the efficiency of the synthesis process and simplifies the purification procedure [18–20].”

-          In order to facilitate the reproduction of results, the material and methods section should give a detailed explanation. Please, briefly describe how the enzymes were produced (even if there is a reference from 10 years ago).

-          Caption of figure 2 still lacks important information. All the abbreviations in the figure should be specified in the caption.

-          The section “results and discussion” still seem a description of methods rather than a discussion comparing with other works and highlighting the novelty of the present work.

-          Table 3, are the authors sure that the accuracy of their results is at the level of the second decimal: in example “148.73±5.06”? I would keep only 1 decimal position.

-          The conclusions are well explained but a brief explanation of the relevance of this study and/or further perspectives would be highly appreciated.

Comments on the Quality of English Language

In my previous report, I strongly recommended English proofreading. The English quality is still not good. Please, can the authors explain how this has been done? In case any artificial intelligence software has been used for this purpose, it should be clearly stated.

Author Response

Q1. The sentences of the abstract (especially at the beginning) seem a list of bullet points. This should be corrected and the sentences should be connected one to another so that the text follows a story. The abstract must present, which is the problem and the current state of the art? What is the goal of this work? How have the authors addressed the problem? What is the main conclusion of the study?

The abstract has been corrected.

Q2. Yet, some new references that have been added to the Reference List are not cited in the text. For example: ref 30, Kaspar et al., 2020, https://doi.org/10.1002/adsc.201901230. Maybe the reference 29 in line 376 should be ref 30?

References 21–33 have been corrected.

Q3. Although the introduction has been improved, substantial changes are still needed. These two sentences say basically the same. Please, modify accordingly: “Currently, research is ongoing to create new nucleosides based on 1,2,4-triazole that have low cytotoxicity and high antiviral activity.” “Currently, ribavirin analogues with substituents on the heterocyclic base are being synthesized and studied for their antitumor and antiviral activities [9,10].”

The first sentence has been changed to: “For this reason, the search for new nucleosides based on 1,2,4-triazole is still relevant.”

Q4. This sentences does not make sense. Please, revise: “Currently, nucleoside analogues are synthesized using various enzymes, increases the efficiency of the synthesis process and simplifies the purification procedure [18–20].”

The sentence has been revised: “Currently, various enzymes are used to produce nucleoside analogues. Enzymatic synthesis increases the efficiency of the synthesis process and simplifies the purification procedure.”

Q5.  In order to facilitate the reproduction of results, the material and methods section should give a detailed explanation. Please, briefly describe how the enzymes were produced (even if there is a reference from 10 years ago).

“Expression and purification of recombinant E. coli PNP and UP” section has been added to Materials and Methods.

Q6. Caption of figure 2 still lacks important information. All the abbreviations in the figure should be specified in the caption.

Abbreviations have been specified in the caption of figure 2.

Q7. The section “results and discussion” still seem a description of methods rather than a discussion comparing with other works and highlighting the novelty of the present work.

The descriptions of methods have been removed.

The following text is added:

1,2,4-Triazole-3-thione (3) has the smallest size of all compounds tested. The presence of only a triazole cycle ring and a thio group does not allow binding to the enzyme active site. Such small compounds have not been reported to be substrates of nucleoside phosphorylases. 5-Phenyl-1,2,4-triazole-3-thione (4) has an additional phenyl group. Compounds containing a five-membered and a six-membered cycle ring are substrates for E. coli purine nucleoside phosphorylase [33], however, no product formation was observed in the reaction with this compound.

New reference [33]: Khandazhinskaya et al. (https://doi.org/10.1039/d1ob01069g) is added.

Q8. Table 3, are the authors sure that the accuracy of their results is at the level of the second decimal: in example “148.73±5.06”? I would keep only 1 decimal position.

Results in Table 3 have been rounded up to one decimal position.

Q9. The conclusions are well explained but a brief explanation of the relevance of this study and/or further perspectives would be highly appreciated.

The following text is added:

“Next, we're going to synthesize and test new 1,2,4-triazole-3-thione derivatives for their antiviral activity on various models.”

Q10. In my previous report, I strongly recommended English proofreading. The English quality is still not good. Please, can the authors explain how this has been done? In case any artificial intelligence software has been used for this purpose, it should be clearly stated.

We didn't use any artificial intelligence software.

The English has been improved.

Reviewer 4 Report

Comments and Suggestions for Authors

No further issues.

Author Response

Thanks for the review.

Round 3

Reviewer 1 Report

Comments and Suggestions for Authors

The paper can now be accepted for publication

Comments on the Quality of English Language

-

Author Response

Thanks for the review.

Reviewer 3 Report

Comments and Suggestions for Authors

The authors have made great improvements in the quality of the presentation of the results of this work. Therefore, I suggest publication in Biomolecules after checking some minor corrections.

-          Line 131-132: Please use the same abbreviations for the chloride: “Tris-HCL” or “NaCl”

-           Line 431: I think there is a typo in the number “337” of this sentence: “The synthesis of ribosides 8, 10, 12 was carried 430 out for 23, 337 and 72 hours, respectively (Table 2).”

-          For the enzymatic transformations, the authors used a ratio 1:2 for the sugar donor (2’-deoxouridine) and a 1:10 ratio for the phosphate. Have the authors checked that these are the most optimal conditions for this reaction? Please, discuss this in relation to previous works, such as https://onlinelibrary.wiley.com/doi/10.1002/adsc.201701005,

-          Line 473, a previous work has also described the use of bulky and hydrophobic nucleobases as substrates for hexameric purine nucleoside phosphorylases: https://chemistry-europe.onlinelibrary.wiley.com/doi/10.1002/cssc.202102030

Author Response

Q1. Line 131-132: Please use the same abbreviations for the chloride: “Tris-HCL” or “NaCl”

Done.

Q2.  Line 431: I think there is a typo in the number “337” of this sentence: “The synthesis of ribosides 8, 10, 12 was carried out for 23, 337 and 72 hours, respectively (Table 2).”

No, the synthesis of compound 10 actually lasted two weeks.

Q3. For the enzymatic transformations, the authors used a ratio 1:2 for the sugar donor (2’-deoxouridine) and a 1:10 ratio for the phosphate. Have the authors checked that these are the most optimal conditions for this reaction? Please, discuss this in relation to previous works, such as https://onlinelibrary.wiley.com/doi/10.1002/adsc.201701005,

The molar ratios of substrate : carbohydrate donor 1:2 and substrate : phosphate 1:10 were used in the test reactions (Table 1). These are suboptimal conditions. The purpose of the test reactions was to determine whether heterocyclic bases are PNP substrates.

The products were then synthesized and isolated. The substrate : donor ratio is given in Table 2. The phosphate concentration has not been optimized. All reactions were carried out at a phosphate concentration of 10 mM. As there was no full optimization, we have not discussed this in the paper.

Q4. Line 473, a previous work has also described the use of bulky and hydrophobic nucleobases as substrates for hexameric purine nucleoside phosphorylases: https://chemistry-europe.onlinelibrary.wiley.com/doi/10.1002/cssc.202102030

The following text is added:

“Compounds with bulky and hydrophobic residues have previously been reported as substrates for bacterial PNP’s [21, 34].”

The reference was added to the paper.